# Depression: A Contributing Factor to the Clinical Course in Myasthenia Gravis Patients

**DOI:** 10.3390/medicina60010056

**Published:** 2023-12-28

**Authors:** Oana Antonia Mihalache, Crisanda Vilciu, Diana-Mihaela Petrescu, Cristian Petrescu, Mihnea Costin Manea, Adela Magdalena Ciobanu, Constantin Alexandru Ciobanu, Ovidiu Popa-Velea, Sorin Riga

**Affiliations:** 1Department of Doctoral Studies, “Carol Davila” University of Medicine and Pharmacy, 020021 Bucharest, Romania; oana.mihalache@drd.umfcd.ro; 2Department of Neurology, “Fundeni” Clinical Institute, 022328 Bucharest, Romania; crisanda.vilciu@umfcd.ro; 3Department of Neurology, “Carol Davila” University of Medicine and Pharmacy, 020021 Bucharest, Romania; diana-mihaela.vlad@rez.umfcd.ro; 4Department of Psychiatry, “Prof. Dr. Alexandru Obregia” Clinical Hospital of Psychiatry, 041914 Bucharest, Romania; cristian.petrescu@drd.umfcd.ro (C.P.); mihnea.manea@live.com (M.C.M.); 5Neuroscience Department, Discipline of Psychiatry, Faculty of Medicine, University of Medicine and Pharmacy “Carol Davila”, 020021 Bucharest, Romania; 6Faculty of Medicine, University of Medicine and Pharmacy “Carol Davila”, 020022 Bucharest, Romania; 7Department of Medical Psychology, Faculty of Medicine, University of Medicine and Pharmacy “Carol Davila”, 050474 Bucharest, Romania; ovidiu.popa-velea@umfcd.ro; 8Department of Stress Research and Prophylaxis, “Prof. Dr. Alexandru Obregia” Clinical Hospital of Psychiatry, 041914 Bucharest, Romania; d_s_riga@yahoo.com; 9Romanian Academy of Medical Sciences, 927180 Bucharest, Romania

**Keywords:** myasthenia gravis, depression, MG-ADL

## Abstract

*Background and Objectives*: The association between myasthenia gravis (MG) and depression is intricate and characterized by bidirectional causality. In this regard, MG can be a contributing factor to depression and, conversely, depression may worsen the symptoms of MG. This study aimed to identify any differences in the progression of the disease among patients with MG who were also diagnosed with depression as compared to those without depression. Our hypothesis focused on the theory that patients with more severe MG symptoms may have a higher likelihood of suffering depression at the same time. *Materials and Methods*: One hundred twenty-two male and female patients (N = 122) aged over 18 with a confirmed diagnosis of autoimmune MG who were admitted to the Neurology II department of Myasthenia Gravis, Clinical Institute Fundeni in Bucharest between January 2019 and December 2020, were included in the study. Patients were assessed at baseline and after six months. The psychiatric assessment of the patients included the Hamilton Depression Rating Scale-17 items (HAM-D), and neurological status was determined with two outcome measures: Quantitative Myasthenia Gravis (QMG) and Myasthenia Gravis Activities of Daily Life (MG-ADL). The patients were divided into two distinct groups as follows: group MG w/dep, which comprised 49 MG patients diagnosed with depressive disorder who were also currently receiving antidepressant medication, and group MG w/o dep, which consisted of 73 patients who did not have depression. *Results*: In our study, 40.16% of the myasthenia gravis (MG) patients exhibited a comorbid diagnosis of depression. Among the MG patients receiving antidepressant treatment, baseline assessments revealed a mean MG-ADL score of 7.73 (SD = 5.05), an average QMG score of 18.40 (SD = 8.61), and a mean Ham-D score of 21.53 (SD = 7.49). After a six-month period, a statistically significant decrease was observed in the MG-ADL (2.92, SD = 1.82), QMG (7.15, SD = 4.46), and Ham-D scores (11.16, SD = 7.49) (*p* < 0.0001). These results suggest a significant correlation between MG severity and elevated HAM-D depression scores. Regarding the MG treatment in the group with depression, at baseline, the mean dose of oral corticosteroids was 45.10 mg (SD = 16.60). Regarding the treatment with pyridostigmine, patients with depression and undergoing antidepressant treatment remained with an increased need for pyridostigmine, 144.49 mg (SD = 51.84), compared to those in the group without depression, 107.67 mg (SD = 55.64, *p* < 0.001). *Conclusions*: Our investigation confirms that the occurrence of depressive symptoms is significantly widespread among individuals diagnosed with MG. Disease severity, along with younger age and higher doses of cortisone, is a significant factor associated with depression in patients with MG. Substantial reductions in MG-ADL and QMG scores were observed within each group after six months, highlighting the effectiveness of MG management. The findings suggest that addressing depressive symptoms in MG patients, in addition to standard MG management, can lead to improved clinical outcomes.

## 1. Introduction

Myasthenia gravis (MG) is well recognized as a prevalent neuromuscular illness characterized by its chronic autoimmune nature [1,2,3]. It is caused by the presence of pathogenic autoantibodies that target and bind specifically to neuromuscular junction components and [2,4,5,6] compromises transmission between nerves and muscles, resulting in the typical weakening of skeletal muscles that worsens with effort [2,4,7].

Approximately 70–85% of individuals with myasthenia gravis (MG) have antibodies targeting the muscle acetylcholine receptor (AChR). These anti-AChR antibodies define the classic seropositive MG phenotype and are highly specific [7,8,9,10]. While approximately 5–6% of patients diagnosed with seronegative acetylcholine receptor (AChR) MG have antibodies against MuSK, lipoprotein receptor-related protein 4 (LRP4) has been found in 2–27% of patients with double-seronegative MG (dSnMG) (where both AChR and MuSK antibodies are absent) [5,8,9,10]. In a minority of MG patients, anti-agrin antibodies can be identified, regardless of AChR, MuSK, or LRP4 antibodies. Some individuals with MG may also possess muscle antibodies that respond to non-junctional antigens like titin, muscular voltage-gated potassium channel complex (Kv1.4), and ryanodine receptor (RyR) [8,9,11]. In turn, a certain percentage of patients remain seronegative, meaning that they do not have any detectable antibodies against these antigens [2,6,9,12].

Acquired myasthenia gravis is considered a rare disease, but its incidence and prevalence have increased in recent years [1,13]. The overall prevalence of MG around the world is 12.4 people per 100,000 population [14], with an estimated annual incidence of 8–10 cases per million person–years [5].

Myasthenia gravis can appear at any age and affects individuals of both sexes [3]. There are two distinct peaks in occurrence, one manifesting at about 30 years of age and the second appearing at around 50. Early-onset MG is more prevalent among younger women; on the other hand, a slightly higher prevalence of males is observed in the older age group [1]. In recent decades, myasthenia gravis incidence among individuals aged 65 and older has increased significantly, with it affecting both men and women [13,15].

The predominant clinical feature of this condition is the fluctuating weakness of the skeletal muscles, although the disease manifests in various ways; most of the time, patients initially present with ocular symptoms [1], and most of them (85%) [5] develop generalized weakness over time. The diagnosis of myasthenia gravis is based on a combination of clinical and paraclinical evaluations. In most cases, the clinical presentation relies on a collection of signs and symptoms such as double vision, drooping eyelids, difficulties with speech and swallowing, respiratory muscle weakness, and fatigue in specific muscles. It typically exclusively involves muscular harm without impacting the sensory or central neurological systems, worsening with physical activity, and manifesting fluctuations in symptoms throughout the day or over long periods [5,12]. As part of the diagnostic process, serum antibody assays are performed as an initial step, focusing on testing for AChR-Ab and MuSK-Ab. Electromyography (EMG) assays are the following diagnostic step in seronegative patients, with them aiding in confirming postsynaptic disorders. Pharmacological testing plays a supportive role in diagnosis; the clinical response to cholinesterase inhibitors aids in affirming the MG diagnosis and may serve as an alternative second-line option to electrodiagnosis. Additionally, requesting computer tomography is essential for thymus screening in patients to aid in managing them [12,16,17]. Myasthenia gravis is characterized by an unpredictable course marked by relapses and occasional remission. The severity of the disease varies considerably between individuals and can even fluctuate within the same person, making it a challenging condition to manage [13].

People diagnosed with myasthenia gravis (MG) struggle with this chronic, life-threatening disease that has a significant impact on various aspects of their lives. An individual’s ability to carry out routine activities such as eating, driving, maintaining personal hygiene, walking, shopping, or cleaning the house is frequently affected. People with MG may also have to change their jobs or sometimes give it up altogether [18,19]. 

Myasthenia gravis may present challenges regarding early identification since it presents symptoms that overlap with psychological conditions such as breathing difficulties, weak facial expression, loss of energy, and tiredness. On the contrary, it is common for psychiatric symptoms that appear during the progression of the condition to be mistakenly referred to as signs of myasthenia, mimicking the relapse of MG and having an essential impact on the evolution of patients [20,21]. Therefore, it is crucial to have a comprehensive understanding of these complex relationships to ensure accurate diagnosis and the timely implementation of appropriate treatments [20,22].

Myasthenia gravis treatment requires continuous medication, and it is crucial to consider both the potential side effects and the mental health implications of specific treatment approaches. When managing myasthenia gravis, corticosteroids are widely used as a first-line immunosuppressive therapy, and their long-term use is linked to mental health disorders [2,23,24]. Patients undergoing multi-drug treatment might have a higher prevalence of psychological problems and a lower quality of life [18,23,25,26]. Another important aspect is the administration of treatment for psychiatric illnesses, which requires careful attention and personalized strategies due to the presence of multiple contraindications in patients with myasthenia gravis for specific classes of psychiatric medications. For example, benzodiazepines, a widely used medication in psychiatry, are associated with worsening myasthenic symptoms; they can cause excessive sedation during the night and respiratory depression. Among the antipsychotic drugs, those with an anticholinergic effect, such as olanzapine, quetiapine, or clozapine, require closer monitoring when administered.

Mental illnesses, in their complex manifestations and various degrees of severity, are a prevalent aspect of the human condition [27]. Diagnosing and treating depression is a primary mental health concern that impacts a substantial proportion of the worldwide population. World Health Organization (WHO) statistics reveal an estimated 3.8% of the population experience depression, with varying rates across different demographic data. Among adults, the prevalence is 5%, with a slight gender difference showing that it affects 4% of men and 6% of women. Furthermore, depression is a concern for older individuals, as approximately 5.7% of adults over 60 struggle with this condition. On a global scale, the 12-month prevalence of this condition fluctuates across countries but averages at around 6% [28]. These numbers collectively highlight the widespread impact of depression, with a staggering 280 million people worldwide facing its challenges. However, the prevalence of depression, when associated with chronic medical conditions, has revealed a striking increase among individuals with one or more chronic diseases. Statistical data reveal a significant overlap, with an average estimate ranging from 9.3% to 25% [29,30]. 

The relationship between chronic diseases such as MG and depression is complex, with bidirectional causality often occurring. On the one hand, chronic diseases can be precursors to depression, imposing a heavy emotional burden on individuals dealing with the challenges of managing long-term health conditions. On the other hand, depression itself can exacerbate the course of MG, potentially impeding effective self-care, compliance with treatment regimens, and overall health outcomes [29].

Cognitive impairment, along with depression and sleep disturbances, is an essential therapeutic concern that must be carefully examined and investigated in patients with MG. The results of recent studies indicate a high occurrence of mild cognitive impairment (MCI) with a complex amnestic cognitive profile in MG. Visuoconstructive/visuospatial skills, followed by verbal short- and long-term memory and selective and divided attention, were observed to be the most frequently affected domains. However, the exact underlying processes responsible for this condition are still being studied [31,32].

Myasthenia gravis is associated with high rates of depression and anxiety, according to accumulating evidence [33,34]. The prevalence of depression in individuals with myasthenia gravis reveals heterogeneity because of the diverse scales and tools used to evaluate patients and the diverse patient populations. The frequency ranges across different continents; for example, Europe has the highest prevalence of depression, at 56%, suggesting a substantial burden of this condition, while Asia reports the lowest prevalence of depression at 28%. Shifting our focus to anxiety, the Americas have the highest prevalence, with 53% of individuals experiencing anxiety-related issues; in contrast, Asia reports the lowest prevalence of anxiety, with just 25% of individuals affected [25]. The prevalence rate is even higher in studies where the selection was made from a tertiary hospital with severe and complicated cases; a study reported that 90% of patients with MG had a psychiatric comorbidity [35]. 

The primary objective of this study was to examine if there are differences in the evolution of the disease in patients diagnosed with depression compared to those without depression. Thus, we hypothesized that patients exhibiting more severe manifestations of MG might exhibit a higher likelihood of experiencing depression; furthermore, the investigation aimed to identify the factors that can have an impact on the evolution of the disease, such as sociodemographic factors, the clinical variables of the disease, and treatment.

During the extensive course of these investigations, one of the goals was to improve our understanding of the complicated diagnostic relationship between MG and depression; the correlation between myasthenia gravis and psychiatric symptoms still remains somewhat ambiguous to date. This pursuit aligns with the urgency of identifying contributing factors and developing new, personalized therapeutic approaches ultimately aimed at improving the quality of life of affected patients.

## 2. Materials and Methods

### 2.1. Study Design and Participants

This study was conducted at the Neurology II department of Myasthenia Gravis, Clinical Institute Fundeni in Bucharest, between January 2019 and December 2020. Patients admitted to the neurology clinic during this period were recruited for participation. One hundred and twenty-two male and female patients (N = 122) aged over 18 with a confirmed diagnosis of autoimmune MG fulfilling the inclusion and exclusion criteria registered for the study were included. 

The diagnosis of MG was confirmed by the current accepted criteria as a typical clinical manifestation and confirmed by the positivity of at least one of the following: demonstration of a pathognomonic decrement in repetitive nerve stimulation and/or increased jitter at single-fiber electromyography, a positive response of the muscular weakness after the administration of anticholinesterase, and the detection of MG-specific antibodies. Every patient with myasthenia gravis underwent screening for thymoma through CT or MRI scans for mediastinum.

Before their inclusion in the study, all patients had to provide written informed consent after the study procedures had been fully explained, following the Declaration of Helsinki and according to the country’s law. Furthermore, measures were taken to ensure their anonymity. The study received approval from the ethics committee of the Fundeni Clinical Institute, Bucharest, Romania. IRB number-57523.

The exclusion criteria of the study were as follows: patients who declined to take part in the study or did not provide informed consent; patients with any missing clinical data; those with a history of psychiatric disorders as specified by the DSM V criteria (including brief bipolar disorder, schizophrenia, and dementia)—patients previously diagnosed with depression and/or anxiety were not excluded—those with a history of substance dependence/abuse as defined by DSM-V, and those diagnosed with corticosteroid-induced psychosis, and patients that did not return for reevaluation after six months. 

The sociodemographic data and clinical characteristics were collected from the patients’ electronic medical records and clinical interviews. We obtained information about gender, age, education status (those with higher education—those who graduated from university, and those with lower education—those who completed primary or secondary school), and residency (rural or urban). Disease-related information contained the first symptoms of MG (ocular, bulbar, and generalized symptoms), age at disease onset, duration of illness, the number of hospitalizations, the number of myasthenic crises in the past, types of autoantibodies detected, along with whether the patient had undergone thymectomy and if the histopathological diagnosis was thymoma.

Patients were assessed at baseline and after six months. During both meetings, the patients underwent clinical evaluations carried out by a neurologist and a psychiatrist under the supervision of the head of the neurology and psychiatry department.

The patients were divided into two distinct groups as follows: group MG w/dep, which comprised 49 MG patients diagnosed with depressive disorder who were also currently receiving antidepressant medication, and group MG w/o dep, which consisted of 73 patients who did not have depression.

### 2.2. Psychiatric Assessment

To confirm or establish the diagnosis of depression, each participant was evaluated by an experienced psychiatrist using a semi-structured interview. In this interview, the participants were assessed for the presence of alcohol/substance abuse, mental retardation, dementia, schizophrenia, schizoaffective disorder, and bipolar disorder, and patients who fulfilled the criteria established by the Diagnostic and Statistical Manual of Mental Disorders, Fifth Edition (DSM-5) [36], were diagnosed with or confirmed as having depressive disorder.

The participants underwent evaluation with the Hamilton Depression Rating Scale-17 items (HAM-D), which measures depression and anxiety symptoms. The Hamilton Depression Rating Scale-17 items (HAM-D) [37] is a frequently used clinician-rated scale for assessing depression. It includes a total of 17 items, with a total possible score of 53. Elevated scores indicate greater levels of depressive symptoms and higher levels of anxiety. Of the scoring scale, 0–7 is considered as being normal, 8–16 suggests mild depression, 17–23 suggests moderate depression, and scores over 24 are indicative of severe depression.

The psychiatrist documented if the patients adhered to the prescribed antidepressant treatment regimen and the specific antidepressant drug that the patient was prescribed. Antidepressants of different classes like selective serotonin reuptake inhibitors (SSRIs): escitalopram and sertraline, selective serotonin and norepinephrine reuptake inhibitors (SNRIs): duloxetine, and atypical antidepressants: mirtazapine and trazodone were the choices of antidepressants used for the patients with MG and depression. The medication was administered following the guidelines and dosage recommendations.

### 2.3. Neurological Assessment

Clinical evaluation of the patients was carried out by two independent neurologists. 

The evaluators noted the disease disability as graded by the Myasthenia Gravis Foundation of America (MGFA) classification [38]. The primary objective of classification is to categorize patients into distinct groups according to the severity of their disease and the specific manifestation of symptoms. The MGFA classification system contains five distinct classes, including pure ocular (class I), mild generalized (class II), moderate generalized (class III), severe generalized (class IV), and intubation/myasthenic crisis (class V). Patients falling under the broad classifications of II, III, and IV are categorized into class A if their symptoms predominantly affect the spinal muscles and class B if they predominantly affect the bulbar region. 

The investigators did not intervene in the choice of treatment, and the results of the scales did not influence the subsequent therapeutic conduct. For the current MG therapies, we took into account the daily dose of pyridostigmine and corticosteroids, the presence of other immunosuppressive agents, and whether the patients were receiving antidepressant treatment or not.

The clinical status was determined with two outcome measurements: 

The Quantitative Myasthenia Gravis (QMG) scale [39] was created to evaluate the severity of the disease and quantify endurance and fatigability while considering the variable characteristics of the condition. It consists of 13 objective items, each scored from 0 (normal) to 3 (most severe), and the total QMG score varies between 0 to 39. The QMG score is composed of the following items: ocular (two items), facial (one item), bulbar (two items), gross motor (six items), axial (one item), and respiratory (one item). It is the primary tool used in initial MG clinical trials and is recommended for use in all prospective clinical trials for evaluating treatment-related outcomes [38]. Some studies suggest a QMG score cut-off of 2 for patients with mild (QMG 0–9) to moderate (QMG 10–16) disease and a cut-off of 3 for those with severe MG (QMG > 16). In alternative meta-analyses, it has been observed that a decrease of 3 points in QMG scores might have even greater significance [40].

The Myasthenia Gravis Activities of Daily Living (MG-ADL) scale [41] is a patient-reported scale used to assess symptoms and functional status in individuals with myasthenia gravis. It includes two components related to daily life activities: the ability to perform tasks such as brushing teeth or combing hair and limitations in standing up from a chair. Additionally, it incorporates six items that evaluate different symptoms associated with myasthenia gravis, including diplopia, ptosis, chewing, swallowing, voice/speech difficulties, and respiratory symptoms. The scoring system assigns a value between 0 and 3 to each item, resulting in a total score ranging from 0 to 24. The MG-ADL scale is an intuitive test that can be easily administered without further training. It is also time-efficient, taking less than 10 min to complete. Furthermore, it is suitable for regular clinical practice and clinical trials [42].

### 2.4. Statistical Analysis

The R program was used for the statistical analysis, version 4.2.3 Copyright (C) 2023 The R Foundation for Statistical Computing, R Core Team (2023). R: A language and environment for statistical computing. R Foundation for Statistical Computing, Vienna, Austria [43].

The primary research endpoints were the MG-ADL and QMG scale values, measured at baseline and after six months.

Using bidirectional paired T-tests, it was determined whether there was a reduction in the two scores for each patient group after six months. Furthermore, the differences between the two scores in the two patient groups were also compared using a bidirectional Welch test. A comparative analysis was conducted between the two patient groups.

The comparative analysis between the two groups involved Welch *t*-tests for continuous variables and chi-squared or Fisher’s exact tests for categorical ones.

The statistical significance was attributed to *p*-values that were less than 0.05.

## 3. Results

### 3.1. Demographic and Clinical Data

The present study has a retrospective observational design and included a total number of 122 patients diagnosed with myasthenia gravis. The age range of participants was 19 to 87 years, with a mean age of 54.3 years (SD = 15.8). In total, 81 (66.4%) study participants were female, while 41 (33.6%) were male. Half of them have rural provenance (*n* = 51; 41.8%), and 76 (62.3%) have lower education. Regarding the initial manifestations of myasthenia gravis, it was observed that ocular symptoms were present in 51 individuals (41.8%), while bulbar symptoms were reported in 36 individuals (29.5%). Additionally, 35 patients (28.7%) experienced spinal symptoms as their first presenting symptom. The MG duration ranged from 1 to 384 months with a mean duration of 80.8 months (SD = 91.4). Overall, 110 (90.2%) patients had positive AChR antibodies, while 12 (9.8%) had positive anti-MuSK antibodies or were seronegative. Out of the forty-nine individuals in the study (40.2%) who had undergone thymectomy, thymomas were detected in 18.9% (*n* = 23) of them. Regarding medication, most patients used acetylcholinesterase inhibitors (98.36%), while corticosteroids were used by 119 (97.54%), and 22 (18.00%) received treatment with an immunosuppressant. Details of the clinical and demographic characteristics are presented in Table 1.

### 3.2. Comparative Analysis between Patients with and without Depression

A comparative analysis between the two groups involved the use of Welch *t*-tests for continuous variables and chi-squared or Fisher’s exact tests for categorical ones.

The comparison of sociodemographic characteristics between the two groups is presented in Table 2. Furthermore, a comparison involving clinical factors is also included in Table 3.

Comparison of the scale values and MG treatment at baseline and after six months between the groups are described in Table 4.

In patients without depression, the difference between MG-ADL values at the study’s outset and six months is statistically significant when using the bidirectional paired T-test: mean difference = 2.27, T-statistic = 6.73, degrees of freedom = 72, *p* < 0.0001, and 95%CI = (1.60 to 2.94).

In patients with depression, the difference between MG-ADL values at the study’s outset and six months is statistically significant when using the bidirectional paired T-test: mean difference = 4.81, T-statistic = 8.43, degrees of freedom = 48, *p* < 0.0001, and 95%CI = (3.66 to 5.94).

Comparing the means of differences between the two groups, they are statistically significant (depression vs. non-depression) when using the bidirectional Welch T-test: mean difference = 2.54, T-statistic = 3.83, degrees of freedom = 80.79, *p* = 0.0002, and 95%CI = (1.22 to 3.86).

In patients without depression, the difference between QMG values at the study’s outset and six months is statistically significant when using the bidirectional paired T-test: mean difference = 6.56, T-statistic = 10.25, degrees of freedom = 72, *p* < 0.0001, and 95%CI = (5.28 to 6.83).

In patients with depression, the difference between QMG values at the study’s outset and six months is statistically significant when using the bidirectional paired T-test: mean difference = 11.24, T-statistic = 11.77, degrees of freedom = 48, *p* < 0.0001, and 95%CI = (9.32 to 13.16).

Comparing the means of differences between the two groups, there are statistically significant results (depression vs. non-depression) when using the bidirectional Welch T-test: mean difference = 4.68, T-statistic = 4.07, degrees of freedom = 88.82, *p* < 0.0001, and 95%CI = (2.39 to 6.96).

The correlation between the two scales was investigated by calculating the Pearson correlation coefficient: at the study’s outset, there was a strong correlation between the two scales: r Pearson = 0.80 and *p* < 0.0001; at six months, the correlation was also strong: r Pearson = 0.61 and *p* < 0.0001.

### 3.3. The Predictors’ Individual Impacts on Myasthenia Gravis Progression, as Indicated by Varying QMG Values

The simple univariate linear regression model shows the influence of each predictor on the evolution of the patients (evolution quantified by the difference in the initial QMG score and the QMG score at six months); the chosen predictors were the statistically significantly different variables in the comparative analysis between the two groups. These data are presented in Table 5.

The table shows the isolated (separate) effects of the predictors on the evolution of myasthenia (different QMG). Depression has an almost identical effect on this model compared to the comparative analysis; besides this effect, other statistically significant impacts on the evolution are also observed: the stage of the disease, the presence of myasthenic crises, the HAM-D difference (initial–6 months), higher doses of cortisone, and higher doses of pyridostigmine.

## 4. Discussion

Our investigation confirms that the occurrence of depressive symptoms is significantly widespread among individuals diagnosed with MG. The findings indicated that 40.16% of MG patients in our cohort were diagnosed with depression. This correlates with the established research findings.

Javad Nadali et al. [25], who summarized 38 studies in a systematic review and meta-analysis on the prevalence of depression and anxiety among myasthenia gravis patients, estimated the percentage of depression at 36%. The research examined a wide range of studies, uncovering variations in the prevalence of these psychological symptoms, with rates of depression ranging from 1% to 76% and anxiety rates ranging from 3% to 71%. Variations in these dissimilarities can be ascribed to the methodological approaches and the use of different screening tools. Despite the wide range of tools available, a challenge arises in distinguishing between somatic symptoms shared by MG and those indicative of depression and anxiety. Symptoms such as fatigue and tiredness are common in both conditions, potentially leading to false results in diagnosis. Therefore, the selection of appropriate screening tools is crucial to accurately identify mood disorders in MG patients.

Yury V. et al. [33], who evaluated 68 patients, reported a frequency of moderate–severe depression of 20.5%. Several factors contribute to this high prevalence, including the unpredictable nature of the disease, social status, and the adverse effects of chronic corticosteroid treatment, all of which may contribute to depressive symptoms in MG patients. Furthermore, the authors suggest that the underlying immunopathology in MG may also contribute to a higher susceptibility to depression with a link between depression and abnormalities in both the innate and adaptive immune systems, suggesting that thymectomy might play a role in reducing the frequency of depressive symptoms by influencing changes in the immune system. Despite this argument, our investigation revealed no statistically significant association between the status of thymectomy and the frequency of depressive symptoms. Moreover, the study found that psychosocial fatigue was the strongest independent predictor associated with higher BDI scores. It is worth mentioning that the study noted a significant overlap between depressive symptoms and fatigue in MG patients, attributed by the authors to various factors, including fatigue being both a symptom of depression and a primary symptom of MG, thus creating a challenge to separate the various directions of causality between depressive symptoms and fatigue given the observational nature of the study and the multidimensional aspects of both symptoms.

The findings of our study suggest that individuals with myasthenia gravis who received treatment with antidepressants tend to be younger, with a mean age of 42.27 years (SD = 14.88), than patients without depression, 58.97 years (SD = 14.67, *p* < 0.001), and with a shorter duration of illness, the first group had a mean duration of illness of 59.55 months (SD = 66.98) compared to 95.03 months (SD = 102.64, *p* = 0.023) in the group of MG patients without depression. This statement supports the underlying premise that the etiology of psychiatric morbidity can be attributed to MG’s persistent and debilitating nature, resulting in limitations across several domains of life [35], diminished quality of life, and significant psychological distress [44]. Additionally, the fluctuating symptoms of myasthenia gravis and the possibility of a myasthenia crisis contribute to increased irritability, tension, and anxiety in patients, increasing their susceptibility to depressive and anxiety disorders [45]. In addition, patients newly diagnosed with MG might suffer an important psychological impact upon learning their diagnosis, thus increasing the chances of developing an anxious-depressive syndrome, compared to chronic patients who are already adapted to the lifestyle changes imposed by this diagnosis and have come to know the warning signs that predict a possible decompensation. Furthermore, some studies have revealed that in the first year of the disease, the severity of MG symptoms is at its highest [46].

This is in accordance with the research conducted by Anca Bogdan et al. [47], wherein a prospective longitudinal cohort study was conducted, including 155 patients diagnosed with MG. The patients underwent clinical evaluation and were requested to complete self-administered questionnaires about disease severity, chronic stress, and depression during both visits. The study demonstrated a prevalence rate of depression of 17% and identified a significant positive association between depression and younger age, earlier disease onset, and higher disease severity. The above data do not coincide with the findings of other studies, for example [48]. In the study of O.I. Kalbus et al. [49] the hypothesis regarding the effect of disease duration on the development of depression was not confirmed, and there were no statistically significant correlations between the Beck score and disease duration.

The study demonstrated a statistically significant correlation between the prevalence of depression and the severity of the disease. This result corresponds with the results of other studies [35,47,48,50]. At the baseline of our evaluation, in the group of MG patients with depression and antidepressive treatment, the MG-ADL score was 7.73 (SD = 5.05), while in the group of MG patients without the diagnosis of depression, the MG-ADL score was 4.30 (SD = 4.84, *p* < 0.001), a statistically significant difference.

The baseline observation that patients with depression who were also receiving antidepressive treatment exhibited higher severity scores raises a significant challenge in assessing whether elevated depression scores may play a causal role in the progression of myasthenia gravis. Given this initial disparity in disease severity, it becomes difficult to discern whether high depression scores actively trigger the course of MG. Deriving evidence from previous research, it is well documented that physical stress can be identified as a contributing factor to relapses in myasthenia gravis [51]. A more nuanced evaluation is warranted to establish a clearer understanding of this relationship, considering the complex interplay between depression, MG progression, and the influence of antidepressive treatment.

The coexistence of myasthenia gravis and depressive disorder raises an intricate challenge. It is intriguing to analyze this coexistence as merely coincidental, particularly since somatic symptoms associated with MG might obscure the accurate assessment of the severity of depression. In other words, the physical symptoms experienced by MG patients may sometimes be mistaken for signs of depression [52,53]. It is essential to consider this intricate relationship carefully and not jump to conclusions about the severity of depression in MG patients, as these individuals may be grappling with complex symptoms that are not immediately evident through traditional diagnostic criteria. A comprehensive, interdisciplinary approach to assessing and addressing their mental health is essential to ensure that MG patients’ physical and emotional requirements are addressed.

In the group of MG patients with antidepressant treatment, at baseline, the mean MG-ADL score was 7.73 (SD = 5.05) with a mean Ham-D score of 21.53 (SD = 7.49), and after six months, the MG-ADL score decreased to 2.92 (SD = 1.82) with a mean Ham-D score of 11.16 (SD = 7.49), with the difference being statistically significant (*p* < 0.0001). The Quantitative Myasthenia Gravis (QMG) score assessment also revealed similar statistically significant variations. Specifically, during the first visit, the average QMG score was 18.40 (SD = 8.61); however, after six months, the average QMG score decreased to 7.15 (SD = 4.46). These data suggest a clear association between the severity of MG and higher HAM-D depression scores. The explanation might reside in the fact that patients facing a more advanced disease stage of MG tend to exhibit elevated depression scores, reflecting the emotional toll that the condition can take. Many of these patients, despite their higher depression scores, have been accurately diagnosed with depression and received suitable therapeutic interventions.

In comparing the two groups, the group of patients with MG and depression showed a notable improvement, as the MG-ADL and QMG scores exhibited a significant decrease. However, when juxtaposed with the scores in non-depressive patients, these scores, though still higher, did not reach statistical significance. This outcome underscores the promising potential of the interventions within the group with depression, indicating a positive trend toward better patient outcomes. Nonetheless, further investigation may be required to determine the factors contributing to the persistent score disparity between the two groups.

Regarding the MG treatment in the group with depression, at baseline, the mean dose of oral corticosteroids was 45.10 mg (SD = 16.60), and the mean dose of oral pyridostigmine was 231.43 mg (SD = 84.45), which is significantly higher than in the group without depression, 28.29 mg (SD = 17.80) for the mean dose of corticosteroids and 175.89 mg (SD = 87.94) for pyridostigmine. No statistically significant differences were observed between the corticosteroid doses on the second visit. However, regarding the treatment with pyridostigmine, patients with depression and antidepressant treatment remained with an increased need for pyridostigmine, 144.49 mg (SD = 51.84), compared to those in the group without depression, 107.67 mg (SD = 55.64, *p* < 0.001). The increasing need for pyridostigmine indicates a concerning aspect of patient management. The observed increase in demand may suggest that patients experience persistent myasthenic weaknesses that their prescribed treatment has not fully addressed or that they are exploring self-medication, which includes the potential for overdosing.

According to guidelines, corticosteroids are the first line of immunotherapy medication for MG treatment [5] and the constant need for medication might also cause psychological stress. The use of corticosteroids has the potential to cause a range of behavioral and mood alterations in individuals [22]. Patients with MG may face a spectrum of conditions, ranging from seemingly mild issues to more severe manifestations, such as psychosis. Notably, insomnia and mood fluctuations, spanning from irritability to varying degrees of depression, are widespread but often poorly documented adverse effects affecting nearly all patients. The underreported nature of these challenges underscores the need for heightened awareness and comprehensive documentation to better address the array of difficulties encountered by those with MG.40, [54,55].

In long-term corticosteroid treatment, one must recognize another factor that may significantly influence patients’ mental health: the treatment’s possible adverse reactions. These reactions can range from weight gain and skin modifications such as acne to distressing gastrointestinal side effects [26,50] and, in some cases, even the development of diabetes or osteoporosis. The emotional and psychological toll of coping with these side effects can be profound, often leading to increased stress and anxiety in patients already dealing with the challenges of their underlying conditions.

As highlighted earlier, the study revealed a notable link between the presence of depression among patients and the occurrence of more severe symptoms of the condition. This correlation led to a heightened necessity for increased doses of cortisone and pyridostigmine among these individuals. The concurrent administration of escalated medication dosages alongside depression therapy emerges as a pivotal element in alleviating the progression of the disease and managing its symptomatic impact on affected individuals. This nuanced approach, addressing both the condition and its mental health implications, plays a crucial role in mitigating the overall severity and manifestation of the disease, offering a potential avenue for more comprehensive treatment strategies. 

Studies have shown a correlation between the dose of cortisone and its adverse effects, suggesting that depressive states may persist due to insufficient dose reductions [18,56]. Remarkably, despite the increased risk of depressive symptoms with higher doses of cortisone, these increased doses play an essential role in significantly improving the course of the disease. As patients progress and their need for cortisone decreases, a notable observation emerges: as cortisone doses decrease, positive effects become apparent on depression scales. This correlation underlines the potential double impact of cortisone, where while higher doses improve the symptoms of the disease, their reduction contributes to the improvement of the associated depressive symptoms. Consequently, the hypothesis highlights the potential importance of dose adjustment, concomitant therapies, and psychiatric evaluations in managing this correlation. However, the complexity of this relationship requires further empirical investigation and comprehensive studies to validate and elucidate the complicated relationship between medication dosage, depressive states, and concurrent disease progression.

Fikret Aysal and colleagues [53], in their research regarding the treatment of patients with myasthenia gravis about symptoms of anxiety and depression, suggested that individuals who received a combination of anticholinesterase and immunosuppressive therapies experienced a higher prevalence of depression and anxiety symptoms than those who were solely prescribed prednisolone. Our study did not show a statistical difference between the patients treated only with corticosteroids and those with an immunosuppressant associated with the treatment.

An additional factor to consider is the potential interactions between different medications, for example, those possessing anticholinergic qualities like antihistamines, tricyclic antidepressants, and trazodone. This medication has the potential to reduce the cholinergic effects of pyridostigmine [54].

In the group without depression, the mean HAM-D score at baseline was 5.10 (SD = 5.33), and at the second visit, it was 4.22 (SD = 5.13). Despite having HAM-D scores below 7, it is crucial to maintain vigilant psychiatric monitoring for these patients. This proactive approach allows healthcare professionals to closely monitor patients’ mental health and well-being, allowing them to keep an eye out for any potential development of depressive disorders. Whether it is due to the progression of their underlying condition or personal factors, early identification, and timely intervention can be instrumental in improving their overall quality of life. This ongoing observation ensures that appropriate care and support are readily available, helping these individuals maintain a higher mental health and well-being standard. Fikret et al. [48], in their cross-sectional study of 42 Myasthenia Gravis patients, explored the connection between anxiety symptoms and depression, disease severity, and treatment methods. The authors suggested that patients experiencing traumatic life experiences may require an extensive psychiatric evaluation. Another study conducted by Yongxiang Yang et al. [57], who evaluated factors that might affect the health-related quality of life (HRQoL) of MG patients in cross-sectional clinical research including 188 successive patients with MG, concluded that the findings from their study strongly advocate for the prioritization of active psychological support for myasthenia gravis patients. Recognizing the impact of anxiety and depression symptoms on this patient population, it becomes evident that addressing their psychological well-being should be a crucial aspect of their care. As such, these results provide valuable insights for shaping effective clinical treatment strategies that not only manage the physical aspects of MG but also emphasize the enhancement of patients’ overall quality of life.

Myasthenic crisis is an essential variable in the progression of the disease. The worsening of this condition, together with the recurrence of myasthenic crisis, presents a recurrent issue, significantly increasing the burden of this disease [58]. Despite remarkable strides in diagnostic techniques and therapeutic interventions for MG management, the occurrence of myasthenic crises persists, with them afflicting approximately 15% to 20% of individuals grappling with generalized MG [59,60,61]. Notably, the biochemical alterations observed in MG patients experiencing respiratory distress during these crises bear a resemblance, to a considerable extent, to the triggers often associated with panic attacks—encompassing factors such as hyperventilation, fluctuations in carbon dioxide levels, and increased sodium lactate [61]. The correlation seen between MG, symptoms of anxiety, and the occurrence of panic attacks suggests a possible connection based on shared physiological pathways. By understanding the intricate relationship between biochemical changes during myasthenic crisis and their connection to anxiety triggers, we can gain essential knowledge that may lead to more precise measures and comprehensive management approaches for individuals who struggle with the complex nature of myasthenia gravis.

## 5. Limitations

The present study possesses several limitations. Despite our extensive analysis involving a substantial number of patients, we cannot assert the representativeness of our cohort. Most participants were affiliated with a national self-help group; thus, our research exclusively focused on Romanian myasthenia gravis patients. Psychiatric disorders, notably depression and anxiety, may present diagnostic challenges when distinguishing them from myasthenia gravis and conversely. This diagnostic ambiguity arises from the presence of shared clinical symptoms and the inherent methodological constraints of patient surveys. Another limitation is the short duration of follow-up and the use of a convenience sample composed of patients undergoing MG treatment at a specialized hospital. Likely due to this bias, one notable effect is the higher prevalence of patients in our cohort who are using immunosuppressant medications.

The strengths of our study are the longitudinal study design, the substantial number of patients enrolled, and the use of validated scales.

## 6. Conclusions

Myasthenia gravis (MG), a chronic autoimmune neuromuscular disease, has significant consequences beyond somatic symptoms. This study highlights the importance of adopting a personalized, interdisciplinary approach to MG patients, recognizing that therapeutic success should encompass various aspects, including socioeconomic, professional, and mental well-being. The interaction between MG and psychiatric disorders is complex, making initial recognition and differentiation challenging, as symptoms commonly associated with myasthenia gravis (MG), such as dyspnea, fatigue, and generalized asthenia, may be mistakenly interpreted as indicators of psychological distress. This overlap can lead to misdiagnosis and unnecessary treatments. Accurately identifying the underlying causes of these symptoms is crucial for adopting the most effective treatments and ensuring the well-being of MG patients with the goal of increasing their overall quality of life. However, limited data are available on the prevalence and causes of these psychiatric symptoms in MG patients. Consequently, more comprehensive studies, including prospective, randomized controlled trials, are needed to improve patient management and enhance their quality of life throughout the course of MG. Lastly, healthcare practitioners must maintain a state of alertness and contemplate the potential presence of myasthenia gravis (MG) when assessing individuals exhibiting apparent psychiatric manifestations to guarantee precise identification and suitable treatment.

## Figures and Tables

**Table 1 medicina-60-00056-t001:** Sample characteristics.

Sample Characteristic	Overall (*n* = 122)
Age	
Mean (SD)	54.3 (15.8)
Gender	
F	81 (66.4%)
M	41 (33.6%)
Education	
Primary and secondary	76 (62.3%)
Superior	46 (37.7%)
Residency	
Rural	51 (41.8%)
Urban	71 (58.2%)
Onset	
Bulbar	36 (29.5%)
Ocular	51 (41.8%)
Spinal	35 (28.7%)
Duration of illness	
Mean (SD)	80.8 (91.4)
Hospitalizations	
Mean (SD)	1.30 (0.768)
Myasthenic crisis	
Mean (SD)	0.180 (0.386)
MGFA Clinical Classification	
IIaMGFA	43 (35.2%)
IIbMGFA	62 (50.8%)
IIIaMGFA	5 (4.1%)
IIIbMGFA	12 (9.8%)
Thymectomy	
Yes	49 (40.2%)
No	73 (59.8%)
Thymoma	
Yes	23 (18.9%)
No	99 (81.1%)
Anti-AChR ab	
Negative	12 (9.8%)
Positive	110 (90.2%)

MGFA—Myasthenia Gravis Foundation of America; Anti-AChR ab—antibodies against acetylcholine receptor.

**Table 2 medicina-60-00056-t002:** Comparison between patients with and without depression—sociodemographic characteristics.

Sociodemographic Characteristic	MG w/dep (*n* = 49)	MG w/o dep (*n* = 73)	*p*-Value ^1^
Age, Mean (SD)	47.27 (14.88)	58.97 (14.67)	<0.001
Gender, *n* (%)			0.081
Female	37 (76)	44 (60)	
Male	12 (24)	29 (40)	
Education, *n* (%)			0.56
Primary and secondary	29 (59)	47 (64)	
Superior	20 (41)	26 (36)	
Residency, *n* (%)			0.19
Rural	24 (49)	27 (37)	
Urban	25 (51)	46 (63)	

^1^ Welch two-sample *t*-test; Pearson’s chi-squared test; Fisher’s exact test; MG w/dep—MG patients with depression; MG w/o dep—MG patients without depression.

**Table 3 medicina-60-00056-t003:** Comparison between patients with and without depression—clinical features of patients.

Clinical Features	MG w/dep	MG w/o dep	*p*-Value ^1^
MGFA Classification, *n* (%)			<0.001
IIa	10 (20)	33 (45)	
IIb	28 (57)	34 (47)	
IIIa	1 (2)	4 (5.5)	
IIIb	10 (20)	2 (2.7)	
Thymectomy, *n* (%)			0.006
Yes	27 (55)	22 (30)	
No	22 (45)	51 (70)	
Thymoma, *n* (%)			0.007
Yes	15 (31)	8 (11)	
No	34 (69)	65 (89)	
Anti-AChR ab, *n* (%)			0.99
Negative	5 (10)	7 (9.6)	
Positive	44 (90)	66 (90)	
Immunosuppressants, *n* (%)			0.045
Yes	13 (27)	9 (12)	
No	36 (73)	64 (88)	

Anti-AChR ab—antibodies against acetylcholine receptor. ^1^ Welch two-sample *t*-test; Pearson’s chi-squared test; Fisher’s exact test.

**Table 4 medicina-60-00056-t004:** Comparison of scale value and MG treatment between groups at baseline and after six months.

Scale/Treatment	MG w/dep	MG w/o dep	*p*-Value ^1^
QMG score			
T0, Mean (SD)	18.40 (8.61)	11.88 (7.66)	<0.001
T1, Mean (SD)	7.15 (4.46)	5.32 (3.85)	0.021
MG-ADL scale			
T0, Mean (SD)	7.73 (5.05)	4.30 (3.84)	<0.001
T1, Mean (SD)	2.92 (1.82)	2.03 (1.76)	0.009
HAM-D scale			
T0, Mean (SD)	21.53 (7.49)	5.10 (5.33)	<0.001
T1, Mean (SD)	11.16 (6.58)	4.22 (5.13)	<0.001
Corticosteroids (mg)			
T0, Mean (SD)	45.10 (16.60)	28.29 (17.80)	<0.001
T1, Mean (SD)	13.06 (6.11)	11.51 (6.33)	0.18
Pyridostigmine (mg)			
T0, Mean (SD)	231.43 (84.85)	175.89 (87.94)	<0.001
T1, Mean (SD)	144.49 (51.84)	107.67 (55.64)	<0.001

T0 = evaluation at baseline; T1 = evaluation after 6 months. ^1^ Welch two-sample *t*-test; Pearson’s chi-squared test; Fisher’s exact test.

**Table 5 medicina-60-00056-t005:** The variable’s specific effects on myasthenia gravis progression.

Variables	N	Beta (95% CI) ^1^	*p*-Value
Age	122	−0.04 (−0.12 to 0.03)	0.235
Duration of illness	122	0.00 (−0.02 to 0.01)	0.465
MGFA Clinical Classification			
IIaMGFA	43	—	
IIbMGFA	62	6.7 (4.5 to 8.8)	<0.001
IIIaMGFA	5	7.5 (2.4 to 13)	0.005
IIIbMGFA	12	8.4 (4.9 to 12)	<0.001
Hospitalizations	122	0.77 (−0.71 to 2.3)	0.310
Myasthenic crisis			
No	100	—	
Yes	22	5.9 (3.2 to 8.7)	<0.001
Difference HAM-D	122	0.37 (0.22 to 0.51)	<0.001
Thymectomy			
Yes	49	—	
No	73	1.0 (−1.3 to 3.3)	0.393
Tymoma			
Yes	23	—	
No	99	−1.5 (−4.4 to 1.4)	0.324
Cortizon 1	122	0.24 (0.19 to 0.28)	<0.001
Immunosuppressants			
Yes	22	—	
No	100	−0.76 (−3.7 to 2.2)	0.614
Pyridostigmine 1	122	0.05 (0.04 to 0.06)	<0.001
Depression			
No	73	—	
Yes	49	4.7 (2.5 to 6.8)	<0.001

^1^ CI = confidence interval.

## Data Availability

All data reported within the article are available in anonymized form from the qualified investigators upon request.

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
