# Peer review of "Depression: A Contributing Factor to the Clinical Course in Myasthenia Gravis Patients"

_medicina, 2023, doi:10.3390/medicina60010056_

Round 1

Reviewer 1 Report (Previous Reviewer 1)

Comments and Suggestions for Authors

Satisfactory

Author Response

Thank you very much for your reviewing. 

Reviewer 2 Report (Previous Reviewer 2)

Comments and Suggestions for Authors

The article is a resubmission. It is enough improved, hence I have no further comments.

Author Response

Thank you very much for your reviewing. 

This manuscript is a resubmission of an earlier submission. The following is a list of the peer review reports and author responses from that submission.

Round 1

Reviewer 1 Report

Comments and Suggestions for Authors

1.     Please review the instructions for authors. A well-formatted manuscript is better for the reviewers to revise the manuscript.

2.     The abstract should be reconstructed. There is a lengthy description of the background. However, the abstract needs a more methodological description. Also, the description of the results could have been better. Please provide numbers and ratios in the description of the results. Moreover, the conclusion is different from the aim of the study.

3.     Aim of the study. How does the aim of the present manuscript differ from others in the literature? What is the global significance of the manuscript? The questions in the aim of the present review are well-known and came from epidemiological studies.

4.     Please avoid markers that are uncommonly written in scientific manuscripts.

5.     Please provide the IRB number for the development of this manuscript. This should be written in the section after the conclusion and in the methodology.

6.     A guideline should be cited for the diagnosis of MG, and the inclusion criteria should be described accordingly.

7.     Exclusion criteria. Do the authors exclude all the individuals with a previous diagnosis of depression/anxiety? If these individuals, how were their psychiatric conditions assessed?

8.     Please describe the statistical analysis.

a.     How was calculated the power of the study?

b.     How was the number of individuals enrolled in the study obtained?

c.     How was the distribution of the variables?

d.     What were the statistical methods applied?

e.     How were the variables chosen?

f.      How were confounding variables' influences assessed?

g.     The description of the study's aim does not correspond to the statistics applied.

9.     Table 4 shows the significance of almost every variable with the development of depression. Please verify the data accordingly.

Author Response

Response to Reviewer 1

  1. Please review the instructions for authors. A well-formatted manuscript is better for the reviewers to revise the manuscript.

Answer:  Thank you for reviewing our manuscript and for your suggestion. We acknowledge the importance of a well-formatted manuscript for ease of review by the esteemed reviewers. We assure you that we have thoroughly reviewed the instructions for authors and are committed to ensuring our manuscript adheres to the recommended formatting guidelines. Your feedback is invaluable, and we are taking proactive steps to enhance the presentation of our work to facilitate a more comprehensive and efficient review process.

  1. The abstract should be reconstructed. There is a lengthy description of the background. However, the abstract needs a more methodological description. Also, the description of the results could have been better. Please provide numbers and ratios in the description of the results. Moreover, the conclusion is different from the aim of the study.

Answer:  Thank you for your comprehensive feedback on our abstract. We acknowledge the need for a more balanced and structured approach, particularly in the methodological description and the presentation of results. We meticulously reworked the abstract to include more specific numerical data and ratios to enhance clarity regarding our findings. Furthermore, we recognize the importance of aligning the conclusion closely with the study's aim. You can see the revised abstract according to the instructions in red.

  1. Aim of the study. How does the aim of the present manuscript differ from others in the literature? What is the global significance of the manuscript? The questions in the aim of the present review are well-known and came from epidemiological studies.

Answer: The unique characteristic of our manuscript's objective is its placement within the context of previous scholarly works. Although the inquiries may derive from established epidemiological studies, our manuscript addresses them from a different viewpoint by incorporating various methodologies or perspectives that provide an original perspective for addressing these concerns.

The study of myasthenia gravis has been limited by a lack of extensive research and undertaken on a small cohort of patients due to the rarity of the disease. The uniqueness of our work lies in its longitudinal design, which sets it apart in a subject that tends to be defined by certain limitations. By adopting a longitudinal structure, we aim to transcend the boundaries of previous research, providing a prolonged and comprehensive understanding of the relationship between depression and myasthenia gravis. Through doing this, we aim to contribute to the progress of knowledge and design more efficient management options for this rare disease.

  1. Please avoid markers that are uncommonly written in scientific manuscripts.

Answer: Thank you for your guidance. We reviewed the paper and tried to avoid using markers or indicators not commonly used in scientific manuscripts, aiming to maintain a standard and consistent format throughout the document.

  1. Please provide the IRB number for the development of this manuscript. This should be written in the section after the conclusion and in the methodology.

Answer: Thank you for highlighting this important aspect. The IRB (Institutional Review Board) number pertinent to the development of this manuscript is now included in both the methodology section and after the conclusion, ensuring comprehensive compliance with ethical standards and providing the necessary transparency regarding the study's approval process.

The study was approved by the approved by the Ethics Committee of the Fundeni Clinical Institute, Bucharest, Romania. IRB number- 57523.

  1. A guideline should be cited for the diagnosis of MG, and the inclusion criteria should be described accordingly.

Answer: Thank you for your insightful suggestion. Our manuscript included a specific guideline for diagnosing myasthenia gravis (MG). We outlined the inclusion criteria, aligning them with the reference guide to provide a clear and comprehensive description of the criteria used in our study.

This information and references were added in the 5th paragraph to The predominant clinical feature of this condition.

The references added in the text are listed below:

[5] N. E. Gilhus, S. Tzartos, A. Evoli, J. Palace, T. M. Burns, and J. J. G. M. Verschuuren, “Myasthenia gravis,” Nat Rev Dis Primers, vol. 5, no. 1, p. 30, May 2019, doi: 10.1038/s41572-019

[13] A. Evoli et al., “Italian recommendations for the diagnosis and treatment of myasthenia gravis,” Neurological Sciences, vol. 40, no. 6, pp. 1111–1124, Jun. 2019, doi: 10.1007/s10072-019-03746-1.-0079-y.

[17] E. de P. Estephan, J. P. S. Baima, and A. A. Zambon, “Myasthenia gravis in clinical practice,” Arq Neuropsiquiatr, vol. 80, no. 5 suppl 1, pp. 257–265, May 2022, doi: 10.1590/0004-282x-anp-2022-s105.

[18] R. T. Rousseff, “Diagnosis of Myasthenia Gravis,” J Clin Med, vol. 10, no. 8, p. 1736, Apr. 2021, doi: 10.3390/jcm10081736

  1. Exclusion criteria. Do the authors exclude all the individuals with a previous diagnosis of depression/anxiety? If these individuals, how were their psychiatric conditions assessed?

Answer: Thank you for your inquiry regarding the exclusion criteria related to individuals with a prior diagnosis of depression or anxiety in our study. Patients with pre-existing diagnoses of depression or anxiety were not excluded. They were assessed by a psychiatrist using a semi-structured interview. In this interview, the participants were assessed for the presence of alcohol/substance abuse, mental retardation, dementia, schizophrenia, schizoaffective disorder, bipolar disorder, and depression as defined by the Diagnostic and Statistical Manual of Mental Disorders, Fifth Edition (DSM-5).

Exclusion criteria and the process for evaluating psychiatric conditions is explicitly detailed in the methodology section of the manuscript to ensure transparency and rigor in our research methodology.

  1. Please describe the statistical analysis.
  2. How was calculated the power of the study?
  3. How was the number of individuals enrolled in the study obtained?
  4. How was the distribution of the variables?
  5. What were the statistical methods applied?
  6. How were the variables chosen?
  7. How were confounding variables' influences assessed?
  8. The description of the study's aim does not correspond to the statistics applied.

Answer

We greatly value your commitment to evaluating our statistical analysis. Your observations and comprehensive examination considerably improve the comprehensiveness and precision of our research. We appreciate your careful examination, which enhances the reliability of our findings.

  1. In this study, the power calculation wasn’t performed. The research methodology was designed and executed without prior power analysis.
  2. During the investigation period from January 2019 to December 2020, we recruited hospitalized patients diagnosed with Myasthenia Gravis. This recruitment was conducted exclusively at the Fundeni Clinical Institute's Neurology Clinic II. The study enrolled patients who fulfilled the established inclusion and exclusion criteria. Our patient selection effort aimed to achieve a cohort size that best represented patients diagnosed with Myasthenia Gravis presenting at a tertiary care center.

c.d. The distribution of variables was assessed using several statistical methods tailored to the nature of the data. Bidirectional paired T-tests were employed to determine changes within each patient group over a six-month period, allowing for the evaluation of any reductions in the two scores for individual groups. Additionally, a bidirectional Welch test was utilized to compare differences between the two scores across the patient groups. For comparative analysis between the groups, Welch t-tests were utilized for continuous variables, while categorical variables underwent assessment using chi-square or Fisher's exact tests. These comprehensive statistical approaches were instrumental in evaluating the distributions of both continuous and categorical variables, providing a robust understanding of the data's distributional characteristics within and across patient groups.

  1. The selection of variables underwent a meticulous process grounded in empirical evidence and clinical relevance. Variables were chosen based on their established significance in previous research related to Myasthenia Gravis and the particular focus of our study.
  2. We made a univariate linear regression model to demonstrate the impact of each predictor on the patients' evolution, which is measured by the difference between the initial QMG score and the QMG score at six months. The selected predictors were the variables that showed statistically significant differences in the comparative analysis between the two groups. While the analyses were adjusted for significant potential confounders, it's crucial to acknowledge the possibility of residual confounding factors that couldn't be entirely accounted for, which might still exert an influence.
  3. Thank you for pointing out this discrepancy. Clarifications and changes were made to ensure that the statistical approaches used coherently aligned with the intended objectives presented for the purpose of the study.

  • Table 4 shows the significance of almost every variable with the development of depression. Please verify the data accordingly.

Answer: Thank you for your diligent review of Table 4. We'll reverify the data to ensure accuracy and precision in depicting the significance of each variable concerning the development of depression. Your attention to detail is greatly appreciated, and we'll thoroughly cross-check the information to guarantee the correctness of the presented data.

Reviewer 2 Report

Comments and Suggestions for Authors

This interesting article explores the impact of depression in MG. The topic is very interesting providing possible new insight in the pathophysiology and management of MG. I have some comments:

1.     Title: “Depression: a precipitating factor for the clinical course in Myasthenia Gravis patients”. The title seems not pertinent with the data. There is no evidence of depression as a precipitating factor. I suggest to present it as a factor associated with high severity of the disease. Indeed, there is no mention with worsening or myasthenic crisis.

2.     A recent study in a cohort of MG patients, demonstrated a positive association between Moderate Cognitive Impairment and disease severity, which in turn correlated with depressive symptomatology and sleep disturbances (Frequency and Correlates of Mild Cognitive Impairment in Myasthenia Gravis. Brain Sci. 2023). In that paper patients with generalized MG had a BDI score of 11 and 97% of patients were treated with steroids. Also, a connection with MCI and sleep disturbances was found. No mention was done on “pseudodementia” and sleep deprivation in MG. The authors should compare their findings with the ones presented.

3.     Table 2 clearly shows a connection between factors associated with high severity of disease (thymectomy, MGFA, immunosuppressants) as well as MGADL and QMG scores. Hence it seems that high severity of MG could be associated with the presence of depression.

4.     “At least 2–5% of MG patients do not have 56 detectable antibodies to a known autoantigen [6].” Ref 6 is from 2014. There are more recent reviews on the diagnosis and management of seronegative myasthenia gravis.  

5.     The study was conducted “between 2019 and 2020”. However, no mention to COVID-19 has been done. Indeed, recent evidence shows a detrimental effect of covid-19 infection in MG patients. Moreover, it is likely that higher rates of depression were connected to pandemic itself and quarantine. Please discuss and cite recent papers facing this important topic.

6.     Line 357 and 374. I strongly agree. Was there an association between the presence of depression and longer disease duration?

·      Methods and results are very clear and well-presented

·      Style and grammar are adequate.

Minor:

Line 54: remove “(ACh)”

Line 508: correct “gra vis”

Author Response

Response to Reviewer 2

This interesting article explores the impact of depression in MG. The topic is very interesting providing possible new insight in the pathophysiology and management of MG. I have some comments:

Thank you for reviewing our manuscript and for your engaging feedback on our article exploring the impact of depression in MG. We're delighted that you find the topic intriguing and potentially insightful for shedding light on the pathophysiology and management of MG. Your perspective contributes significantly to the depth and value of our research, and we're grateful for your thoughtful engagement with our study.

  1. Title: “Depression: a precipitating factor for the clinical course in Myasthenia Gravis patients”. The title seems not pertinent with the data. There is no evidence of depression as a precipitating factor. I suggest to present it as a factor associated with high severity of the disease. Indeed, there is no mention with worsening or myasthenic crisis.

Answer: Thank you for your thoughtful input regarding the title of our study, "Depression: a precipitating factor for the clinical course in Myasthenia Gravis patients." Your observation is valuable, and we appreciate your suggestion to refine the title to better align with the findings.

The revised title, "Depression: a precipitating contributing factor for the clinical course in Myasthenia Gravis patients," aims to more accurately capture the essence of our study's findings.

  1. A recent study in a cohort of MG patients, demonstrated a positive association between Moderate Cognitive Impairment and disease severity, which in turn correlated with depressive symptomatology and sleep disturbances (Frequency and Correlates of Mild Cognitive Impairment in Myasthenia Gravis. Brain Sci. 2023). In that paper patients with generalized MG had a BDI score of 11 and 97% of patients were treated with steroids. Also, a connection with MCI and sleep disturbances was found. No mention was done on “pseudodementia” and sleep deprivation in MG. The authors should compare their findings with the ones presented.

Answer: Thank you for highlighting the insightful findings from the recent study on MG patients, particularly the positive association between Moderate Cognitive Impairment (MCI), disease severity, and their correlation with depressive symptoms and sleep disturbances. Your suggestion to compare these findings with the results of our study, particularly regarding topics such as "pseudodementia" and sleep deprivation in MG, is sensible. In our research, we've referenced the findings from the article you mentioned, providing a foundational understanding of the positive association between Moderate Cognitive Impairment (MCI), disease severity, and their correlation with depressive symptoms and sleep disturbances in MG patients. Building upon the insights gleaned from this cited article, we aim to explore and expand upon these relationships, potentially uncovering further nuances and implications for the management and understanding of Myasthenia Gravis.

  1. Table 2 clearly shows a connection between factors associated with high severity of disease (thymectomy, MGFA, immunosuppressants) as well as MGADL and QMG scores. Hence it seems that high severity of MG could be associated with the presence of depression.

Answer: We appreciate your insightful clarification regarding Table 2, which shows the strong association between characteristics associated with the increased severity of Myasthenia Gravis (MG), such as MGFA classification, and the utilization of immunosuppressants. These characteristics demonstrate significant relationships with MGADL and QMG scores, suggesting their impact on the severity of the condition. Your observation on a possible correlation between the severity of MG and the existence of depression is precise. This highlights a substantial opportunity for additional investigation into how disease severity and depressive symptoms interact in MG patients. Your excellent observation facilitates further research into the comprehensive understanding of MG and its related manifestations.

  1. “At least 2–5% of MG patients do not have 56 detectable antibodies to a known autoantigen [6].” Ref 6 is from 2014. There are more recent reviews on the diagnosis and management of seronegative myasthenia gravis.  

Answer: Thank you for highlighting the evolving research context in diagnosing and treating seronegative Myasthenia Gravis (MG). Incorporating the latest insights from recent reviews will undoubtedly enrich and enhance the comprehensiveness of our discussion on seronegative MG in the context of our research. Your emphasis on more current sources aligns with our commitment to presenting the most contemporary and comprehensive information available on Myasthenia Gravis.

This information and references were added in the second paragraph.

The references added in the text are listed below:

[7] C. Farmakidis, M. Pasnoor, M. M. Dimachkie, and R. J. Barohn, “Treatment of Myasthenia Gravis,” Neurol Clin, vol. 36, no. 2, pp. 311–337, May 2018, doi: 10.1016/j.ncl.2018.01.011.

[8] Z. Yu et al., “Characterization of LRP4/Agrin Antibodies From a Patient With Myasthenia Gravis,” Neurology, vol. 97, no. 10, pp. e975–e987, Sep. 2021, doi: 10.1212/WNL.0000000000012463.

[9] Y. Li, Y. Peng, and H. Yang, “Serological diagnosis of myasthenia gravis and its clinical significance,” Ann Transl Med, vol. 11, no. 7, pp. 290–290, Apr. 2023, doi: 10.21037/atm-19-363.

[10] K. Lazaridis and S. J. Tzartos, “Autoantibody Specificities in Myasthenia Gravis; Implications for Improved Diagnostics and Therapeutics,” Front Immunol, vol. 11, Feb. 2020, doi: 10.3389/fimmu.2020.00212.

[11] C. Vinciguerra et al., “Diagnosis and Management of Seronegative Myasthenia Gravis: Lights and Shadows,” Brain Sci, vol. 13, no. 9, p. 1286, Sep. 2023, doi: 10.3390/bainsci13091286.

[12] R. Yamashita et al., “Anti-MuSK Positive Myasthenia Gravis with Anti-Lrp4 and Anti-titin Antibodies,” Internal Medicine, vol. 60, no. 1, pp. 137–140, Jan. 2021, doi: 10.2169/internalmedicine.4957-20.

  1. The study was conducted “between 2019 and 2020”. However, no mention to COVID-19 has been done. Indeed, recent evidence shows a detrimental effect of covid-19 infection in MG patients. Moreover, it is likely that higher rates of depression were connected to pandemic itself and quarantine. Please discuss and cite recent papers facing this important topic.

Answer: Indeed, your point regarding the impact of COVID-19 on MG patients and its potential influence on depression rates during the pandemic is significant. However, it's important to note that the scope of this current study was set before the pandemic. The focus of this research did not encompass the COVID-19 factor. Acknowledging the potential implications of the pandemic on depression rates and the well-being of MG patients is vital. We recognize the importance of this topic and intend to explore it as a part of future research initiatives to comprehensively address the multifaceted impacts of external factors, such as COVID-19, on MG patients' mental health.

  1. Line 357 and 374. I strongly agree. Was there an association between the presence of depression and longer disease duration?

Answer: Thank you for pointing out the references at lines 357 and 374.

Certainly, our study hinted at an exciting trend suggesting a potential association between increased depressive symptoms and a shorter duration of illness among Myasthenia Gravis (MG) patients. Contrary to the anticipated hypothesis regarding depression and longer disease duration, our findings pointed in a different direction, highlighting a possible link between heightened depressive symptoms and a shorter duration of illness in this patient cohort. This unexpected association prompts further investigation and deeper analysis to understand better the intricate dynamics between disease duration and the prevalence or severity of depression in individuals living with MG.

  • Methods and results are very clear and well-presented
  • Style and grammar are adequate.

Thank you for your positive feedback regarding the clarity and presentation of the methods and results in our research. We strive to ensure precision and coherence in our communication, and it's encouraging to hear that our efforts have resulted in a clear and well-presented portrayal of the study's methodology and findings. Additionally, we appreciate your acknowledgment of the adequacy in style and grammar, as maintaining readability and correctness is essential in effectively conveying our research. Your feedback motivates us to uphold these standards of clarity and accuracy in our work.

Minor:

Line 54: remove “(ACh)”

Line 508: correct “gra vis”

Answer: Thank you for your specific suggestion regarding Line 54 and line 508. We have made the necessary changes. See the changes in red.
